# Mandibular Brown Tumor as a Result of Secondary Hyperparathyroidism: A Case Report with 5 Years Follow-Up and Review of the Literature

**DOI:** 10.3390/ijerph18147370

**Published:** 2021-07-09

**Authors:** Veronika Shavlokhova, Benjamin Goeppert, Matthias M. Gaida, Babak Saravi, Frederic Weichel, Andreas Vollmer, Michael Vollmer, Christian Freudlsperger, Christian Mertens, Jürgen Hoffmann

**Affiliations:** 1Department of Oral and Cranio-Maxillofacial Surgery, University Hospital Heidelberg, 69120 Heidelberg, Germany; Frederic.Weichel@med.uni-heidelberg.de (F.W.); andr.vollmer@gmail.com (A.V.); Michael.Vollmer@med.uni-heidelberg.de (M.V.); christian.freudlsperger@med.uni-heidelberg.de (C.F.); christian.mertens@med.uni-heidelberg.de (C.M.); Juergen.Hoffmann@med.uni-heidelberg.de (J.H.); 2Institute of Pathology, University Hospital Heidelberg, 69120 Heidelberg, Germany; benjamin.goeppert@med.uni-heidelberg.de (B.G.); Matthias.Gaida@unimedizin-mainz.de (M.M.G.); 3Institute of Pathology, University Medical Center Mainz, Johannes Gutenberg-Universität Mainz, 55131 Mainz, Germany; 4Department of Orthopedics and Trauma Surgery, Medical Centre-Albert-Ludwigs-University of Freiburg, Faculty of Medicine, Albert-Ludwigs-University of Freiburg, 79106 Freiburg, Germany; babak.saravi@jupiter.uni-freiburg.de

**Keywords:** brown tumor, secondary hyperparathyroidism, jaw, radical resection, microvascular reconstruction

## Abstract

Background: Brown tumor is a rare skeletal manifestation of secondary hyperparathyroidism. Although diagnosis of the disease is increasingly seen in early stages due to improved screening techniques, some patients still present in a progressed disease stage. The treatment depends on tumor mass and varies from a conservative approach with supportive parathyroidectomy to extensive surgical resection with subsequent reconstruction. Case presentation: We report a case of extensive mandibular brown tumor in a patient with a history of systemic lupus erythematosus, chronic kidney disease, and secondary hyperparathyroidism. Following radical resection of the affected bone, reconstruction could be successfully performed using a free flap. Conclusions: There were no signs of recurrence during five years of close follow-up. Increased awareness and multidisciplinary follow-ups could allow early diagnosis and prevent the need for radical therapeutical approaches.

## 1. Introduction

Hyperparathyroidism (HPT) is an endocrine disorder characterized by elevated secretion of parathyroid hormone, which can result in imbalanced osteoclast activity. In rare cases, such a disturbance in bone turnover can result in skeletal manifestations of HPT. Brown tumor is one of the skeletal manifestations of HPT. The lesion is characterized by non-neoplastic reactive tissue, extensive osteoclastic bone resorption with osteoclast-like multinucleated giant cells, osseous microfractures, and hemorrhage as well as hemosiderin depositions. The term “*brown tumor*” refers to an accumulation of hemosiderin pigment, giving the lesion its macroscopically brown appearance [1].

Incidence of primary hyperparathyroidism (PHPT) varies and ranges between 0.4–21.6 cases per 100,000 person-years [2]. The most common reason for primary hyperparathyroidism is adenoma (80%), parathyroid hyperplasia (10–25%), multiple adenomas (5%) and in rare cases, parathyroid carcinoma (<1%) [3]. Although osteoporosis is the most prevalent skeletal manifestation (prevalence: 39–62.9%), approximately 20% of patients will eventually develop cystic lesions [1]. In contrast, secondary hyperparathyroidism (SHPT) is a common complication of late-stage renal diseases. According to the International Burden of Chronic Kidney Disease and Secondary Hyperparathyroidism, it can affect up to 54% of hemodialysis patients [4]. SHPT is associated with increased exposure of parathyroid tissues to chronic hypercalcemia [1].

In recent years, thanks to improved screening techniques and more frequent early detection of hyperparathyroidism, brown tumors have become a rarity. The general prevalence of brown tumors is low (approximately 0.1%); however, it is increasingly found in PHPT and SHPT patients [5]. The pathology can be found in up to 4.5% and 1.5–1.7% of patients with PHPT and SHPT, respectively [5]. Lesions can affect any skeletal structure and can be solitary or multiple, but they are more commonly localized along long bones, the pelvis, ribs, and clavicles [1]. Notably, involvement of the facial region is extremely rare and is found in only 4.5% of brown tumor cases [6]. On plain radiographs, brown tumors present as well-defined lytic ballooning lesions with a thin peripheral bone shell and several internal thin bony bridges. Although sclerotic bone lesions are most common, lytic lesions are also seen [7]. Multiple lesions can imitate bone metastases in radiological imaging [7]. As the facial region is less often involved and secondary hyperparathyroidism is generally associated with a lower incidence of brown tumors than primary hyperparathyroidism, physicians may not be aware of this pathology in these cases. This can lead to diagnosis at a later stage with a greater tumor mass presenting.

Here, we report a rare case of an extensive brown tumor with manifestation in the mandible of a patient with secondary HPT, who also had systemic lupus erythematosus and a history of chronic renal failure. Further, we provide a review of the available relevant evidence, including other case reports published to date.

## 2. Case Presentation

A 41-year-old female presented to our hospital with suspicious swelling of the lower jaw. She reported that the swelling on the right side of her face had slowly increased in size over the past 4–6 weeks. An intraoral examination revealed a hard swelling that extended from the right premolars to the ascending ramus. The buccal mucosa and lingual mandibular plates were expanded, leading to a narrowed buccal sulcus. Tooth 45 was loose and showed attenuated pulp sensory responses upon thermal stimulation.

A panoramic X-ray revealed an osteolytic lesion, partial loss of cortical bone, and early root resorption of teeth 45 and 46 (Figure 1).

A CT scan of the head and neck region was performed to determine the anterior-posterior expansion of the lesion and carry out virtual planning of reconstruction. The lesion involved the body of the right mandible with caudal displacement of the mandibular canal. The resulting subtotal bony destruction involved both buccal and lingual cortical bone and measured approximately 38 × 17 × 26 mm (Figure 2 and Figure 3). No lymphadenopathy was seen. Additionally, the hands, skull, shoulder, knee, and pelvis region were screened radiologically to exclude further lytic bony manifestation.

The patient had a history of systemic lupus erythematosus and suffered for approximately 10 years from chronic kidney disease and osteoporosis. She had no family history of hypocalcemia or endocrine disorders and received regular nephrology screening. Associated high blood pressure was controlled with an angiotensin-converting enzyme inhibitor to slow progression of renal failure. An elevated parathyroid hormone (PTH) level of 200.8 pmol/L was measured at the time of presentation (normal range: 1.2–4.5 pmol/L). The biochemical preoperative investigations showed hypocalcemia (2.08 mmol/L; normal range: 2.11–2.59 mmol/L), Vitamin D levels of 20 µg/L (normal range: >20 µg/L), creatinine at 3 mg/dL (normal range: 0.5–0.9 mg/dL), and calcitonin levels of 11 pg/mL (normal range: <6.4 pg/mL) (Table 1).

The initial clinical diagnosis was a giant cell lesion, based on the clinical examination and considering the localization in the right mandible, tumorous swelling, a lack of localized increased temperature, and normal coloration. Clinical differential diagnosis included ameloblastoma, SHPT brown tumor, and aneurysmal bone cyst. Thus, an incisional biopsy was taken. Microscopically, the lesion consisted of plump fibroblasts and numerous osteoclast-like giant cells surrounded by reactive bone formation (Figure 4). Furthermore, focal deposits of hemosiderin and small blood vessels were found. The histomorphological structure was consistent with a brown tumor, while differential diagnoses included giant cell lesions such as giant cell tumor of bone, central giant cell reparative granuloma, low-grade osteosarcoma, and primary or secondary aneurysmal bone cysts. Additional molecular diagnostics were performed for differential diagnosis evaluations. They revealed no rearrangement of the USP6 gene (USP6-Split-FISH) and no amplification of the MDM-2 gene (MDM-2-CISH). Subsequently, a brown tumor associated with secondary hyperthyroidism was diagnosed based on clinical examination and histopathology.

Because of the extent of the tumor and the resulting functional problems for the patient, extensive surgical removal was planned. Notably, a limited amount of remaining intact bone was present in the mandibular region. Figure 2 clearly shows the extent of the tumor with subtotal bony destruction.

Therapeutic options were discussed among the patient and nephrologist, with the conclusion that surgery was most appropriate, owing to the patient’s young age and her stable renal function.

Radical resection with immediate reconstruction with a deep circumflex iliac artery (DCIA) flap was performed. Virtual 3D planning was carried out to better plan the resection and reconstruction and ensure a short operation time (Figure 5).

The DCIA was harvested in standard fashion following the method originally described by Taylor et al. [8]. Hypotensive anesthesia was used during harvesting. Prophylaxis antibiosis included ampicillin and sulbactam (3gr, Unacid, Pfizer). For orientation, the anterior superior iliac spine was marked and the femoral vessels in the inguinal crease were palpated and identified. The incision was marked one finger’s width above and parallel to the inguinal ligament. With the help of cauterization, iliacus muscle was mobilized, followed by osteotomy of the iliac crest with an oscillating saw.

After isolating the bone flap, the pedicle was prepared and divided after the recipient area was ready. A suction drain was placed in the donor area and wound closure was performed in multiple layers. Bupivacaine infiltration (AstraZeneca) was provided at the donor site to minimize postoperative pain. After wound closure, Steri-Strips (3M Health Care, Neuss, Germany) and Cosmopor E (15 × 8 cm; Paul Hartmann, Heidenheim, Germany) were used to cover the wound and a. pressure bandage was applied. The postoperative status was satisfying. After 7–10 days, the nephrological situation improved to preoperative levels after initial temporary acute kidney injury due to tubular necrosis. The intraoral wound healed with no signs of infection. The donor site morbidity was examined preoperatively and during the perioperative period with a specific orthopedic IOWA Hip-Score [9]. Overall, there were significant functional restrictions and limitation of motion and muscle strength for up to one month after surgery. We found no signs of postoperative nerve impairment.

Clinical examination, including patient nephrological status, was carried out immediately post-operation, and five years of regular follow-ups revealed satisfying results with no evidence of recurrence. A post-operative cone-beam computed tomography scan with a 3D reconstruction of the patient showed no signs of tumor leftovers. The follow-up intervention one year postoperatively included a successful implant-prosthetic rehabilitation of the affected area (Figure 6).

## 3. Literature Review

A review of the literature identified a total of 23 patients with secondary HPT resulting in brown tumors of the facial bones (Table 2). Overall, 19 studies published between 2004–2019 were found.

Countries reporting cases of SHPT were Serbia, Greece, Spain, Brazil, India, Bulgaria, Iran, USA, Turkey, Belgium, Indonesia, and China. Most reports of brown tumors associated with secondary hyperparathyroidism came from India. Of the 23 patients identified, 16 were females, and 7 were males (2.3:1). The mean age was 36.3 years, ranging from 12 to 70 years. Overall, 16/25 (64%) brown tumor lesions occurred in the mandible, whereas 9/25 (36%) were found in the maxilla. Reported tumor sizes ranged between 2 × 2 cm to 6 × 7 cm. The majority of patients were treated surgically. One case was successfully treated conservatively with 60,000 IU Vitamin D3 once a week and calcium (500 mg) twice daily [24]. Follow-up data were available for 10 reported cases and ranged from 5 months to 7 years. None of these studies report signs of tumor recurrence during follow-up.

## 4. Discussion

Secondary HPT is most often a result of chronic renal disease with decreased calcium levels and increased PTH secretion. Considering the increasing availability of hemodialysis and diagnostic capabilities, prevalence may increase in the future [4]. Pathogenesis involves a lack of calcitriol synthesis, an active form of Vitamin D, due to attenuated renal function. This leads to decreased calcium levels, which consequently increases serum phosphate levels [29]. Phosphate can directly stimulate PTH secretion in parathyroid glands. Furthermore, fibroblast growth factor 23 (FGF-23) is involved in SHPT pathogenesis, as it inhibits the synthesis and secretion of PTH [30]. This can decrease otherwise elevated phosphate levels [31]. Notably, FGF-23 is reduced in patients with SHPT [31]. Attenuated calcium-sensing receptor (CaSR) and Vitamin D receptor (VDR) expression that is already existent in the early stages of hyperparathyroidism affects parathyroid cell response to calcium and calcitriol [29]. A positive feedback loop of increased phosphate, reduced calcium/calcitriol, and reduced response of parathyroid cells to inhibitory stimuli leads to an increase of proliferative activity and finally parathyroid hyperplasia [29]. Molecular mechanisms leading to skeletal manifestation have not yet been thoroughly researched.

Osteitis fibrosa cystica, often used as a synonym for brown tumors, is a rare skeletal manifestation and was first described by Recklinghausen in 1891 [32]. As a result of the overproduction of PTH in PHPT or SHPT, osteoblasts will increase RANKL expression, which binds to its corresponding RANK receptor on osteoclasts and promotes osteoclast activity. PTH also reduces osteoprotegerin (PG) levels, preventing RANKL and RANK interactions and thus inhibiting bone resorption. Consequently, the lesion can be misdiagnosed as an aneurysmal bone cyst or metastatic carcinoma when multiple lesions exist. An examination of blood markers in suspected cases is therefore essential to point to the correct diagnosis.

Brown tumors are reported to be more common in the mandibles of young female patients, with the highest incidence between the third and sixth decades [1,23]. This is in accordance with the findings of our literature review and case characteristics. The most prevalent lesion sites in the facial region are the jaw, palatinal bone, and nasal and orbital sinuses [20].

Treatment of brown tumors depends on the extent of the tumor mass and associated functional problems. Close control and adjustment of calcium, phosphate, and 1,25-dihydroxy Vitamin D blood levels in patients with chronic renal disease are crucial. The first steps in therapy include normalization of calcium and phosphate levels and thus a multimodal therapeutic approach. Administration of calcitriol or Vitamin D receptor activators (VDRAs) is often sufficient to control SHPT and attenuate tumor mass growth [31]. Unresponsive severe HPT may require removal of parathyroids, leading to regression of brown tumors in many cases [4].

Surgical removal of the tumoral masses may be necessary for patients with inadequate response to conservative therapy and parathyroidectomy. This is often required due to esthetic complaints and functional problems affecting chewing and swallowing [23]. Furthermore, brown tumors may cause pathological fractures and neurological issues. Extensive cases can lead to dysphagia and represent a severe medical condition if not adequately treated. Biopsy of suspected lesions with subsequent enucleation and curettage, including the surrounding healthy bone, is the standard therapy for small, solitary localized lesions. More extensive lesions, including multiple lesion spots and lesions with unclear margins, are subjected to marginal or segmental resections. As there are no reported cases with recurrence of tumor mass after resection or malignant transformations, a more minimal-invasive surgical resection is the therapy of choice. However, in certain cases with large progressed lesions, radical resection will still be necessary.

Due to subtotal local destruction of the mandible and high risk of local complications, such as pathological fracture, radical surgical excision of the brown tumor was undertaken in our patient. In such cases requiring extensive resections, surgical reconstruction can be challenging due to possible complications associated with prolonged general anesthesia in patients with chronic renal diseases. These complications include an elevation of blood urea level, creatinine, decreased erythropoietin production, and reduction of acid, salt, and water excretion, promoting anemia, acidosis, hyperkalemia, hypertension, edema, and platelet dysfunction with resulting bleeding.

Successful reconstructive strategies should aim to optimize functional and esthetic outcomes while reducing donor site morbidity. Furthermore, reconstruction of the jaw should allow future dental rehabilitation to maintain the full function of the masticatory system. The type of defect, along with the patient characteristics, are generally the main factors in making the choice of therapeutic approach. In contrast to historical strategies of mandibular reconstruction, focusing on a staged or delayed approached to account for the recurrence of disease, today’s strategies are increasingly aimed at one-stage therapy with immediate reconstruction [33]. Immediate reconstructions are also associated with significantly improved health-related quality of life outcomes and are preferred by patients [33]. Reconstruction modalities for mandibular reconstruction range from alloplasts, free grafts with cortical bone, pedicled osteomyocutanous flaps, and pedicled myocutaneous non-vascularized bone grafts, which are reserved for minor defects where sufficient soft tissue is available.

Vascularized bone grafts are required for large defects with soft tissue loss when healing independent of an available recipient bed is necessary. We aimed for a reliable and long-lasting reconstruction of large bone loss and face shape, creating a surface for dental restorations and providing a stable wound that would not result in an orocutaneous fistula. Mandibular reconstruction with vascularized flaps is associated with early bone union and less resorption. Further, it provides a generous amount of tissue compared to reconstruction with metal plates or non-vascularized bone grafts. Microvascular free flaps are technically challenging but are a reliable and stable therapeutic option with good outcomes even for large defects involving the anterior mandible. Vascularized free flaps include fibular free flaps, radial forearm free flaps, scapula free flaps, anterior thigh flap, pectoralis major myocutaneous flap, metatarsus osteocutaneous flap, metatarsus osteocutanous flap, and the DCIA flap.

Notably, the present case was the first with a brown tumor associated with secondary hyperparathyroidism to be treated by 3D planning with a subsequent microvascular reconstruction of the jaw for functional rehabilitation. The iliac crest bone is an excellent option for reconstructing the mandible, as its cancellous bone structure allows future dental rehabilitation with dental implants. Furthermore, the shape reconstruction of the mandible, including the anterior mandible, can be effectively planned with 3D planning techniques using the iliac crest due to their similarity in shape. Another advantage of DCIA flaps is the simultaneous reconstruction of intraoral mucosal defects caused by radical resection using the internal oblique muscle, which is more reliable than skin flaps [33]. One oft-reported limitation of the technique is the donor site morbidity. Rogers et al. report that donor site morbidity caused by reconstructions following head and neck cancer significantly affected the patient’s health-related quality of life [34]. However, they did not observe a difference in donor site morbidity between fibula free flaps or DCIA flaps.

Considering the young age of our patient and relatively stable nephrological status, we decided to reconstruct the large mandible defect with a DCIA flap as a one-stage procedure. The iliac crest free flap provided a sufficient amount of cortical and cancellous substance to reconstruct the height of the native mandible. Further, it provided the basis for successful dental rehabilitation with implants.

## 5. Conclusions

Brown tumors are rare osteolytic bone lesions that may occur in patients with primary, secondary, or tertiary hyperparathyroidism. Surgical intervention may be required in cases with extensive lesions. We found no reported recurrences after surgical resection in the cases described in the literature to date. Radical resection and subsequent reconstruction with a microvascular iliac crest graft were successfully performed in our patient presenting with a large progressed mandibular brown tumor. We found no signs of recurrence during five years of close follow-up. Increased awareness among physicians and multidisciplinary follow-ups could allow early diagnosis and prevent radical therapeutical approaches in patients with hyperparathyroidism.

## Figures and Tables

**Figure 1 ijerph-18-07370-f001:**
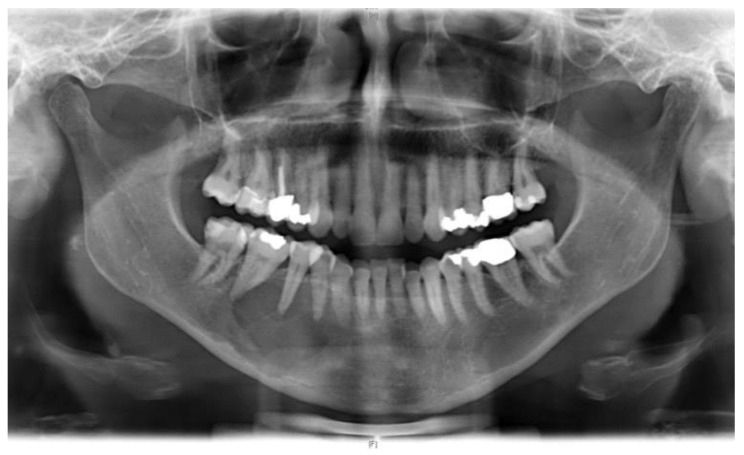
Panoramic X-ray showing an osteolytic lesion on the right side of the mandible. The lesion is associated with root and cortical bone resorption.

**Figure 2 ijerph-18-07370-f002:**
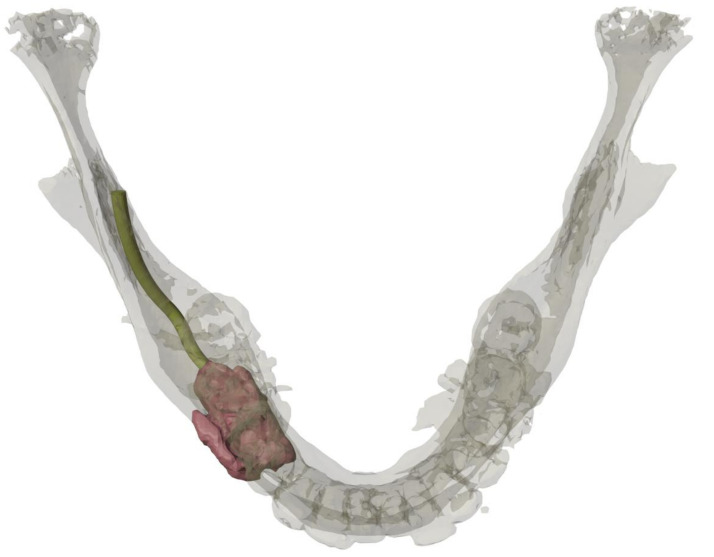
CT reconstruction of the mandible showing the nerve course and the osteolytic lesion on the right side of the mandible.

**Figure 3 ijerph-18-07370-f003:**
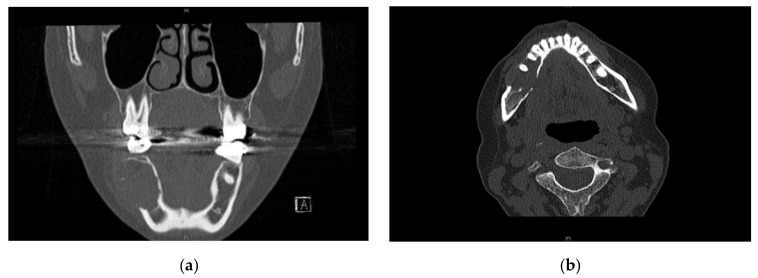
CT scans showing expanded cortical bone on the right side of the mandible. (**a**) coronal plane, (**b**) axial plane.

**Figure 4 ijerph-18-07370-f004:**
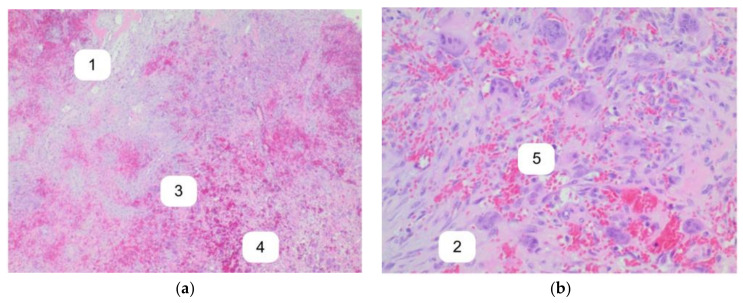
H&E stains show a giant cell lesion consisting of plump fibroblasts (1) and numerous osteoclast-like giant cells (2) surrounded by reactive bone formation (3). Extensive hemorrhage, focal deposits of hemosiderin (4), and small blood vessels (5) are present. Atypical cells are not found. Original magnification: (**a**) 12.5× and (**b**) 200×.

**Figure 5 ijerph-18-07370-f005:**
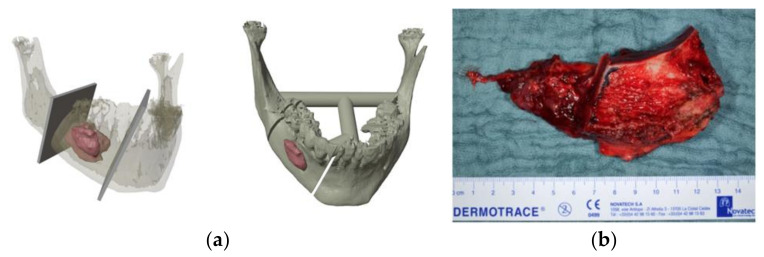
Illustration of (**a**) preoperative 3D osteotomy planning and (**b**) intraoperative image of DCIA flap for reconstruction.

**Figure 6 ijerph-18-07370-f006:**
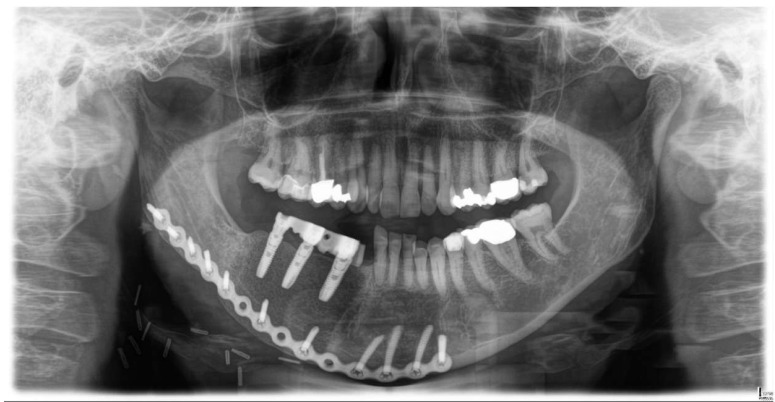
X-ray showing the postoperative situation, one year after surgery. Implant-prosthetic rehabilitation was successfully performed in the affected area.

**Table 1 ijerph-18-07370-t001:** Patient blood values before surgery and follow-up.

Parameters	Pre-Operative	Post-Operative	6-Month Follow-Up	12-Month Follow-Up	24-Month Follow-Up	5-Year Follow-Up	Normal Range
Calcium	2.08 mmol/L	2.27 mmol/L	2.3 mmol/L	2.27 mmol/L	2.2 mmol/L	2.32 mmol/L	2.11–2.59 mmol/L
Phosphate	0.78 mmol/L	1.4 mmol/L	1.21 mmol/L	1.13 mmol/L	1.6 mmol/L	1.39 mmol/L	0.84–1.45 mmol/L
PTH	200.80 pmol/L	7.74 pmol/L	12.3 pmol/L	9.57 pmol/L	10.8 pmol/L	13.6 pmol/L	1.2–4.5 pmol/L
Vitamin D	20 µg/L	72 µg/L	65 µg/L	63 µg/L	56 µg/L	82 µg/L	20–70 µg/L
Creatinine	3 mg/dL	2.6 mg/dL	2.76 mg/dL	2.15 mg/dL	2.58 mg/dL	2.96 mg/dL	0.5–0.9 mg/dL

PTH, parathyroid hormone.

**Table 2 ijerph-18-07370-t002:** Reported cases of brown tumor associated with secondary hyperparathyroidism.

Author and Reference	Year	Country	Sex	Age	Tumor Location	Tumor Size	Local Treatment	Follow-Up
Jović et al. [10]	2004	Serbia	M	25	Maxilla	-	Surgical resection	-
Triantafillidou et al. [11]	2006	Greece	F	21	Mandible	-	Surgical resection	7 years
			F	70	Mandible	-	Surgical resection	7 years
			F	68	Mandible	-	Surgical resection	7 years
Pérez-Guillermo et al. [12]	2006	Spain	M	61	Maxilla	3 × 2 cm	Surgical resection	-
Leal et al. [13]	2006	Brazil	F	31	Maxilla	-	Surgical resection	-
Jakubowski [14]	2011	USA	M	49	Mandible	8 × 2 × 4 cm	-	-
Preeti P. Nair [15]	2011	India	F	35	Mandible	2.5 × 2.7 cm	Conservative (Vitamin D Supplementation)	-
Thomas et al. [16]	2011	India	F	27	Mandible	4 × 6 cm	Surgical resection	9 months (no recurrence)
Praveen and Thriveni [17]	2012	India	F	21	Maxilla and mandible	6 × 7 cm; 2 × 2 cm; 4 × 2 cm	Surgical resection	3 years (no recurrence)
Arunkumar et al. [18]	2012	India	F	12	Mandible	-	Surgical resection	6 months (no recurrence)
Pechalova and Poriazova [19]	2013	Bulgaria	M	19	Mandible	-	Surgical resection	-
			F	49	Maxilla	-	Surgical resection	-
Verma Pradhuman [20]	2014	India	F	31	Mandible	6 × 5 cm	-	-
Jafari-Pozve Nasim [21]	2014	Iran	M	28	Maxilla	-	Parathyroidectomy	-
Mohammed Qaisi [22]	2015	USA	M	43	Mandible	5 × 7 cm	Surgical resection	-
Queiroz et al. [23]	2016	Brazil	F	53	Mandible	-	Surgical resection	8 months (no recurrence)
Özgür Can et al. [24]	2016	Turkey	F	30	Mandible/maxilla	-	Surgical resection	6 months (no recurrence)
Brabyn Philip [25]	2017	Spain	F	42	Maxilla	-	-	-
Aerden Thomas [26]	2018	Belgium	M	32	Mandible	-	-	-
Kartini Diani [27]	2018	Indonesia	F	26	Mandible	3 × 4 cm	Surgical resection	7 months (no recurrence)
			F	31	Maxilla	4 × 3 cm	Subtotal parathyroidectomy	5 months (no recurrence)
Xu Weibo [28]	2019	China	F	30	Mandible	-	Surgical resection	-

## Data Availability

The datasets acquired and/or analyzed during the current study are available from the corresponding author upon reasonable request.

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
