# Peer review of "Mandibular Brown Tumor as a Result of Secondary Hyperparathyroidism: A Case Report with 5 Years Follow-Up and Review of the Literature"

_ijerph, 2021, doi:10.3390/ijerph18147370_

Round 1

Reviewer 1 Report

Dear Authors,

The case presented here reports the resective and reconstructive management of a Mandibular brown tumor as a result of secondary hyperparathyroidism with excellent results at 5 years.

Congratulations to the authors for the management of the case and for the description of the same. Case presentation, discussion as well as literature review are well reported.

However, a more detailed description of the surgical procedures is missing, a precise description of what a deep circumflex iliac artery (DCIA) flap is and how to perform it is missing.

3D planning images would be helpful for understanding. Intraoperative photographs would be very useful if not essential.

Best regards,

Author Response

Thank you for your constructive comments. We highly appreciate the time and effort you have invested in our manuscript. We have tried to address each of your recommendations as precisely as possible.

Reviewer 2 Report

The manuscript should be subjected to a major revision.

General considerations: some sentences should be corrected and rephrased to allow a better English accuracy (ex: line n° 32-34, 46-49, 50-53, 70-72, 77, 104, 108-110, 145-149). The grammatical form and verbal tenses should be revised (ex: lines n° 35, 39, 43, 45, 56, 64, 67, 70, 82, 84, 91, 114, 121, 132).

Introduction: Better analyze the epidemiological and etiological knowledge since the references ranging between 1962 and 2004 years.

Case presentation: Familial medical history should be reported before preoperative blood values list (lines 84-85). Referred to "Table 1" after the presentation of the preoperative blood levels (line 84). Alternatively, the table should be placed later the post-operative follow-up (line 114). Anyway, the reference to "Table 1" should be indicated, to allow the readers to consult the postoperative blood parameters. Better analyze the intraoral examination to clarify the indications and the choice of a preoperative incisional biopsy. Specify the clinical diagnostic hypothesis. The further diagnostic investigations should be performed for the differential diagnosis. It is not appropriate to refer to the lack of collaboration between the clinician and the pathologist. At lines 105-107 is sufficient indicated the limited presence of intact mandibular bone. Figure 4. Correct the punctuation after "Figure 4".

Discussion: The literature review should be specified in a separate paragraph before the discussion. At least, the number of articles, the period time and the countries included in the review, must be indicated. The discussion must deal thoroughly the therapeutic and rehabilitative modalities with their indications, complications, and aesthetic-functional implications. A concise and precise overview of the disease etiopathogenesis should be included. Improve the comparison between the literature data and the presented case report. Lines 146-148 should be deleted. Lines 170-177 should be included in “case presentation”.

References: It is necessary a more thoroughly research of the most recent bibliographic data.

Author Response

(The authors gave the same response as above.)

Reviewer 3 Report

This is a very interesting and well-documented case report of a rare clinical situation. Moreover, authors performed a focused review of the previously reported cases, and presented all information in a comprehensive and didactic form. References are well cited.

In this reviewer opinion, there are only minor adjustments that can be performed to increase que quality of this paper:
1- Figure 4 needs some labeling and marks in the picture to depict what is described in the legend. If a higher quality of the image provided can be acquired would be a surplus;

2- P4L111 – The composition of the flap can be clarified at this point, to allow the reader a full understanding of all the sequential approach;

3- Figure 5 - how many years post-intervention is this radiographic follow-up. Please indicate in the legend;

4- Virtual 3D planning of the resection and reconstruction could be described (software, printing models to guide bone donation area?, splinting???).

Author Response

(The authors gave the same response as above.)

Round 2

Reviewer 2 Report

The authors made all required changes